# Learning Interpretable Decision Rule Sets: A Submodular Optimization Approach

**Fan Yang, Kai He, Linxiao Yang, Hongxia Du, Jingbang Yang, Bo Yang, Liang Sun**
DAMO Academy, Alibaba Group, Hangzhou, China
{fanyang.yf,kai.he,linxiao.ylx,hongxia.dhx,jingbang.yjb,muhai.yb,liang.sun}
@alibaba-inc.com

## Abstract

Rule sets are highly interpretable logical models in which the predicates for decision are expressed in disjunctive normal form (DNF, OR-of-ANDs), or, equivalently, the overall model comprises an unordered collection of if-then decision rules. In this paper, we consider a submodular optimization based approach for learning rule sets. The learning problem is framed as a subset selection task in which a subset of all possible rules needs to be selected to form an accurate and interpretable rule set. We employ an objective function that exhibits submodularity and thus is amenable to submodular optimization techniques. To overcome the difficulty arose from dealing with the exponential-sized ground set of rules, the subproblem of searching a rule is casted as another subset selection task that asks for a subset of features. We show it is possible to write the induced objective function for the subproblem as a difference of two submodular (DS) functions to make it approximately solvable by DS optimization algorithms. Overall, the proposed approach is simple, scalable, and likely to be benefited from further research on submodular optimization. Experiments on real datasets demonstrate the effectiveness of our method.

## 1 Introduction

Interpretability is becoming one of the key considerations when deploying machine learning models to high-stake decision-making scenarios. Black box models, such as random forests and deep neural networks, may achieve impressive prediction accuracy in practice, but it is often difficult to understand the mechanisms about how their predictions are made. Moreover, it has been widely known that machine learning models are susceptible to spurious correlations, which makes them not robust to distribution shifts or adversarial attacks and leads to misleading predictions. Black box models are particularly problematic here because they are hard to audit and to diagnose.

The development of inherently interpretable models is a longstanding attempt towards interpretable and trustable machine learning [42, 43]. In this paper, we consider decision rule sets for binary classification, in which if an example is tested true for a Boolean condition expressed in disjunctive normal form (DNF), then a predefined label is predicted for it. For instance, a rule set for determining whether a patient with community-acquired pneumonia should be hospitalized may be "IF (IsChild = True AND OxygenSaturation < 0.9) OR (IsChild = True AND SOB = True) OR (IsAdult = True AND CURB65 > 2) THEN Yes". Rule sets are particularly suited for tabular data that contain mixed-type features and exhibit complex high-order feature interactions. When compared to other rule-based models such as decision trees and decision lists, the simpler combinatorial structure of rule sets makes them easier to interpret and to learn from data.

The learning of rule sets has attracted interest from various research communities over the past few decades. Two interplaying subset selection tasks constitute the major challenges in rule set learning. First, the construction of a rule requires choosing a subset of all features. Then a subset of all possible

35th Conference on Neural Information Processing Systems (NeurIPS 2021).

rules has to be selected to form a rule set. In early algorithms from machine learning community, such as FOIL [41], CN2 [12] and RIPPER [13], usually a greedy sequential covering strategy is utilized, in which at each stage a rule is generated from uncovered examples using heuristics and newly covered examples are removed. Associative classification techniques [33, 50] developed by data mining community take a different two-stage strategy, in which a large set of rules are first generated via association rule mining and then a rule set is constructed by ranking and pruning. Both these early approaches lack a global objective that guides the generation of rules and optimizes the interpretability of produced rule set.

This paper presents a new submodular optimization perspective on nearly optimal rule set learning. We employ an objective function that simultaneously optimizes accuracy and interpretability. We show that given a ground set of rules, this objective is a regularized submodular function [27], for which algorithms with strong approximation guarantees have been developed recently [23]. Instead of using a pre-mined pool of rules as the ground set, we take an on-the-fly approach in which the ground set consists of all possible rules and elements are picked from it one at a time through solving an optimization subproblem. When the subproblem is always solved to optimality, our learning algorithm enjoys the algorithmic guarantees for regularized submodular maximization. For most real cases the global optimum of the subproblem is impractical to reach, therefore we propose an efficient approximate algorithm based on iterative local combinatorial search.

Our contributions are summarized as follows: **(i)** We formulate interpretable rule set learning as a regularized submodular maximization problem which is amenable to approximate algorithm with strong guarantees. **(ii)** We discover an intuitive difference of submodular (DS) decomposition for the induced rule construction subobjective, based on which an iterative refinement algorithm is proposed to solve the subproblem approximately for large datasets. **(iii)** We conduct a comprehensive experimental evaluation, demonstrating the proposed approach is competitive against state-of-the-arts in both predictive performance and interpretability.

The remainder of this paper is organized as follows: Related work is summarized in Section 2. Section 3 introduces the problem formulation. Section 4 presents the algorithmic details. Experimental evaluation is reported in Section 5. We discuss the limitations of this work in Section 6 and conclude the paper with Section 7.

## 2    Related work

To discover rules from training data, heuristic sequential covering or separate-and-conquer strategy is utilized by almost all early algorithms [41, 12, 13], in which the interpretability of final rule set is not explicitly controlled. In recent years, with the renewed interest in interpretable rule-based models, the rule set learning problem was revisited under modern optimization lens and several new algorithms were developed [22, 49, 28, 35, 48, 51]. These algorithms mainly follow a two-stage paradigm originating from early associative classification methods [33, 50], in which the learning problem is divided into two separate steps: **rule generation** and **rule selection**. For example, Lakkaraju et al. [28] frame the task of constructing a interpretable rule set from pre-mined rules as a submodular optimization problem and solve it approximately with a local search algorithm. In general, such two-stage paradigm suffers from the drawback that some essential rules may be filtered out prematurely.

Dash et al. [14] formulate the rule set learning problem as an integer program (IP) and solve its linear programming relaxation using the column generation (CG) framework [15, 17], in which rules are instead generated on the fly by iteratively solving a IP subproblem. In this way they bridge rule generation and rule selection together as a unified optimization procedure and thus avoid the drawbacks of pre-mining. We contribute a similar framework from a different submodular optimization perspective. Interestingly, subproblems in both frameworks share the same form (6), which asks the solution to cover more positively weighted samples and less negatively weighted samples. This task shares some similarities with the subgroup discovery [6] problem in data mining, for which exhaustive search [7, 31], heuristic beam search [30, 46] and sampling [9, 10] are commonly utilized. Eckstein and Goldberg [17] designed a specialized branch-and-bound method for the closely related maximum monomial agreement (MMA) problem. To the best of our knowledge, we are the first to study this task from a DS optimization perspective.

Our work is also related to optimization based learning of other rule models, such as optimal decision lists [3, 4] and optimal decision trees [24, 34, 8, 39, 2, 47, 52]. The learning of decision lists and trees is inherently much harder because their combinatorial structures are more complex than rule sets.

## 3 Problem formulation

In this work, we consider binary classification based on a set of interpretable binary features. Categorical features can be easily binarized via one-hot encoding, and numerical features may be binarized through bucketing or comparisons like "Age $\geq 18$". Techniques such as one-versus-one are available for transforming multiclass classification to binary classification.

Suppose we are given the training data $\{(\mathbf{x}_i, y_i)\}_{i=1}^n$, in which each $\mathbf{x}_i = (x_{i,1}, \ldots, x_{i,d}) \in \{0, 1\}^d$ is a binary feature vector associated with a label $y_i \in \{0, 1\}$. The goal is to learn an interpretable rule set model $F : \{0, 1\}^d \to \{0, 1\}$ to predict the label. A rule set is a Boolean classifier of the form "IF $P(\mathbf{x})$ THEN $y = 1$ ELSE $y = 0$", in which $P(\mathbf{x})$ is a predicate restricted in disjunctive normal form (DNF) and treats raw features $\{x_j\}_{j=1}^d$ as its variables. For simplicity, the use of not ($\neg$) operator in DNF is disabled because equivalent expressivity can be achieved by creating augmented features $\{\neg x_j\}_{j=1}^d$. Then a concrete $P(\mathbf{x})$ is described as a set $\mathcal{S} \subseteq 2^{[d]}$, where each element $\mathcal{R} \in \mathcal{S}$ is a subset of $[d] := \{1, \ldots, d\}$. The mapping from a $\mathcal{S}$ to a DNF is given by $P_{\mathcal{S}}(\mathbf{x}) := \bigvee_{\mathcal{R} \in \mathcal{S}} \bigwedge_{j \in \mathcal{R}} x_j$. We call $\mathcal{S}$ a rule set and call each element $\mathcal{R}$ of it a rule. A sample $\mathbf{x}$ is **covered** by a rule $\mathcal{R}$ if $P_{\{\mathcal{R}\}}(\mathbf{x})$ is true. It is now clear that a rule set classifier can be simplified to $y = P_{\mathcal{S}}(\mathbf{x})$, i.e., it predicts 1 iff $\mathbf{x}$ is covered by at least one of the rules.

To learn a rule set from data, we consider minimizing the empirical error while penalizing the complexity by solving the optimization problem:

$$\min_{\mathcal{S} \subseteq 2^{[d]}} \sum_{i=1}^n l\left(P_{\mathcal{S}}(\mathbf{x}_i), y_i\right) + \Omega(\mathcal{S}) \tag{1}$$

However, even under the simplest setting where $l(\cdot, \cdot)$ is the 0-1 loss and $\Omega(\cdot)$ is the cardinality function, this problem is still hard to work with. To proceed, we replace $P_{\mathcal{S}}$ with a surrogate model $\tilde{P}_{\mathcal{S}} : \{0, 1\}^d \to \mathbb{N}_0$ defined as $\tilde{P}_{\mathcal{S}}(\mathbf{x}) = \sum_{\mathcal{R} \in \mathcal{S}} \bigwedge_{j \in \mathcal{R}} x_j$. Then a flexible objective function that balances accuracy and interpretability is chosen to be:

$$L(\mathcal{S}) = \sum_{i=1}^n l_{\boldsymbol{\beta}}\left(\tilde{P}_{\mathcal{S}}(\mathbf{x}_i), y_i\right) + \lambda \sum_{\mathcal{R} \in \mathcal{S}} |\mathcal{R}| \tag{2}$$

where $l_{\boldsymbol{\beta}}(\cdot, \cdot)$ is a loss function with hyperparameters $\boldsymbol{\beta} = (\beta_0, \beta_1, \beta_2) \in \mathbb{R}_+^3$:

$$l_{\boldsymbol{\beta}}(\hat{y}, y) = \begin{cases} \beta_0 \hat{y} & \text{if } y = 0 \\ \beta_1(y - \hat{y}) & \text{if } y = 1 \wedge \hat{y} \leq 1 \\ \beta_2(\hat{y} - y) & \text{if } y = 1 \wedge \hat{y} > 1 \end{cases}$$

With a proper choice of hyperparameters, interpretability of the minimizer for this objective comes from two aspects: sparsity and diversity. Sparsity is obvious because of the complexity regularization term. Diversity is encouraged by the loss function through penalizing the overlap of rules. $\tilde{P}_{\mathcal{S}}(\mathbf{x}) > 1$ means that $\mathbf{x}$ is covered by more than one of the rules, and in this case the objective function will receive a penalty. Small overlap among rules benefits interpretability because it makes the decision boundaries more clear and enables people to inspect the learned rules individually.

It may not be immediately clear that $L(\mathcal{S})$ is a supermodular function and thus $-L(\mathcal{S})$ is a submodular function. To show this, we introduce more notations. Let $\mathcal{X}^+ = \{i | y_i = 1\}$ and $\mathcal{X}^- = \{i | y_i = 0\}$ be the set of positive and negative samples, respectively. Let $\mathcal{X}_{\mathcal{S}}^+ = \{i | i \in \mathcal{X}^+ \wedge P_{\mathcal{S}}(\mathbf{x}_i)\}$ and $\mathcal{X}_{\mathcal{S}}^- = \{i | i \in \mathcal{X}^- \wedge P_{\mathcal{S}}(\mathbf{x}_i)\}$ be the set of positive and negative samples covered by $\mathcal{S}$, respectively. Then we have

$$L(\mathcal{S}) = \beta_0 \sum_{\mathcal{R} \in \mathcal{S}} \left|\mathcal{X}_{\{\mathcal{R}\}}^-\right| + \beta_1 \left|\mathcal{X}^+ \setminus \mathcal{X}_{\mathcal{S}}^+\right| + \beta_2 \left(\sum_{\mathcal{R} \in \mathcal{S}} \left|\mathcal{X}_{\{\mathcal{R}\}}^+\right| - \left|\mathcal{X}_{\mathcal{S}}^+\right|\right) + \lambda \sum_{\mathcal{R} \in \mathcal{S}} |\mathcal{R}|$$

Ignoring the weights, then the first term is the miscoverage of negative samples, the second term is the number of uncovered positive samples, the third term is the overlap in positive samples, and the last term is the size of rules. After reorganizing the terms, we arrive at

$$L(\mathcal{S}) = \beta_1 \left| \mathcal{X}^+ \right| - (\beta_1 + \beta_2) \left| \mathcal{X}_{\mathcal{S}}^+ \right| + \sum_{\mathcal{R} \in \mathcal{S}} \beta_0 \left| \mathcal{X}_{\{\mathcal{R}\}}^- \right| + \beta_2 \left| \mathcal{X}_{\{\mathcal{R}\}}^+ \right| + \lambda |\mathcal{R}| \tag{3}$$

Then the submodularity immediately follows.

**Proposition 1.** $L(\mathcal{S})$ *is a supermodular function and* $-L(\mathcal{S})$ *is a submodular function.*

*Proof.* The first term in (3) is a constant. The second term is a negative coverage, while coverage functions are well-known to be submodular. The third term sums over elements of $\mathcal{S}$ and thus is modular. A submodular function minus a modular function is still submodular, therefore $-L(\mathcal{S})$ is submodular and $L(\mathcal{S})$ is supermodular. $\qquad \square$

We show in the Appendix that several design choices for the original objective in (1) may reduce to $L(\mathcal{S})$ through upper bounding surrogates, which justifies the generality of this objective function. This objective function also generalizes the Hamming loss used in previous work [44, 14].

## 4 Algorithms

We consider the algorithms for minimizing $L(\mathcal{S})$ in this section. Define $g(\mathcal{S}) = (\beta_1 + \beta_2) \left| \mathcal{X}_{\mathcal{S}}^+ \right|$ as the revenue of a rule set and $c(\mathcal{R}) = \beta_0 \left| \mathcal{X}_{\{\mathcal{R}\}}^- \right| + \beta_2 \left| \mathcal{X}_{\{\mathcal{R}\}}^+ \right| + \lambda |\mathcal{R}|$ as the cost of a rule. Let $V(\mathcal{S}) = g(\mathcal{S}) - \sum_{\mathcal{R} \in \mathcal{S}} c(\mathcal{R})$ be the profit of a rule set, which is equal to $-L(\mathcal{S})$ up to a constant. Then minimizing $L(\mathcal{S})$ is equivalent to maximizing $V(\mathcal{S})$. For the sake of practicality, we further put a cardinality constraint $|\mathcal{S}| \leq K$ on this optimization problem, which limits the number of rules in the rule set. Then

$$\max_{\mathcal{S} \subseteq 2^{[d]}, |\mathcal{S}| \leq K} V(\mathcal{S}) \tag{4}$$

is an instance of cardinality constrained submodular maximization problem. For non-negative monotone submodular functions, the greedy algorithm achieves a $1 - 1/e \approx 0.632$ approximation ratio [38], which is the best approximation possible for this problem. However, in our case $V(\mathcal{S})$ is neither non-negative nor monotone. It is possible to apply non-monotone submodular maximization algorithms [11, 20] to our problem without care of negativeness, but doing this makes their approximation guarantees invalid.

Fortunately, $V(\mathcal{S})$ is the difference between a non-negative monotone submodular part $g(\mathcal{S})$ and a non-negative modular part $\sum_{\mathcal{R} \in \mathcal{S}} c(\mathcal{R})$. Set functions with such structure, recently named *regularized* submodular functions [27], have been shown to be amenable to maximization procedures with strong approximation guarantees [45, 23, 18, 19].

### 4.1 Regularized submodular maximization

We apply the distorted greedy algorithm proposed by Harshaw et al. [23] to approximate maximization of $V(\mathcal{S})$. As illustrated in Algorithm 1, at each iteration, a rule maximizing the marginal gain in a distorted objective is added to the rule set if its gain is positive. The distorted objective is adjusted in a way such that initially higher importance is placed on the cost term, and then the importance is gradually shifted back to the revenue term. By doing this, following approximation guarantee is obtained.

**Proposition 2.** *[23] Algorithm 1 returns a rule set* $\mathcal{S}$ *of cardinality at most* $K$. *If the maximization subproblem at line 5 is solved to optimality, then*

$$V(\mathcal{S}) = g(\mathcal{S}) - \sum_{\mathcal{R} \in \mathcal{S}} c(\mathcal{R}) \geq (1 - 1/e)g(OPT) - \sum_{\mathcal{R} \in OPT} c(\mathcal{R})$$

*where OPT is the optimal solution to problem* (4).

---

**Algorithm 1** Rule set learning

---

1 **Input:** Training data $\{(\mathbf{x}_i, y_i)\}_{i=1}^n$, hyperparameters $(\boldsymbol{\beta}, \lambda)$, cardinality $K$
2 Initialize $\mathcal{S} \leftarrow \emptyset$
3 **for** $k = 1$ **to** $K$ **do**
4     Define $v_k(\mathcal{R}) = (1 - 1/K)^{K-k} g(\mathcal{R}|\mathcal{S}) - c(\mathcal{R})$         /* $g(\mathcal{R}|\mathcal{S}) \coloneqq g(\mathcal{S} \cup \{\mathcal{R}\}) - g(\mathcal{S})$ */
5     Solve $\mathcal{R}^\star \leftarrow \arg\max_{\mathcal{R} \subseteq [d]} v_k(\mathcal{R})$
6     **if** $v_k(\mathcal{R}^\star) > 0$ **then** $\mathcal{S} \leftarrow \mathcal{S} \cup \{\mathcal{R}^\star\}$ **end if**
7 **end for**
8 **Output:** $\mathcal{S}$

---

In addition, we provide an algorithm in the Appendix for further refining the output of Algorithm 1 if possible. In each step, that algorithm tries to improve the objective through adding, removing, or replacing a rule. The remaining question is how to solve the subproblem of marginal gain maximization, which requires search over the subsets of $[d]$. Naive exhaustive enumeration only works for small $d$, say, $d < 20$. For datasets with numerical features or high-cardinality categorical features, the preprocessing step may easily produce hundreds or thousands of binary features, making exhaustive enumeration impossible. In such cases, approximate solution should be considered and the guarantee for Algorithm 1 will no longer hold. However, we observe in practice that Algorithm 1 will still work satisfactorily if good enough solutions to the subproblem are found. To this end, approximate rather than exact method is considered in this work.

## 4.2   Solving the subproblem

We approximately solve the marginal gain maximization subproblem through iteratively refining a solution with local search until no further improvement can be made. Our local search procedure relies on a decomposable structure of the subproblem to find good solutions quickly. Objective functions for the subproblem are of the general form:

$$
\begin{aligned}
v(\mathcal{R}|\mathcal{S}; \alpha) &= \alpha g(\mathcal{R}|\mathcal{S}) - c(\mathcal{R}) \\
&= \alpha(\beta_1 + \beta_2) \left| \mathcal{X}_{\{\mathcal{R}\}}^+ \setminus \mathcal{X}_{\mathcal{S}}^+ \right| - \beta_0 \left| \mathcal{X}_{\{\mathcal{R}\}}^- \right| - \beta_2 \left| \mathcal{X}_{\{\mathcal{R}\}}^+ \right| - \lambda |\mathcal{R}| \\
&= [\alpha(\beta_1 + \beta_2) - \beta_2] \left| \mathcal{X}_{\{\mathcal{R}\}}^+ \setminus \mathcal{X}_{\mathcal{S}}^+ \right| - \beta_0 \left| \mathcal{X}_{\{\mathcal{R}\}}^- \right| - \beta_2 \left| \mathcal{X}_{\{\mathcal{R}\}}^+ \cap \mathcal{X}_{\mathcal{S}}^+ \right| - \lambda |\mathcal{R}|
\end{aligned}
\tag{5}
$$

where $\alpha \in (1/e, 1]$ is a multiplier on the marginal gain of $g$ set by Algorithm 1. For notational simplicity, denote $\omega_+ = \alpha(\beta_1 + \beta_2) - \beta_2$. Notice that samples are partitioned into three subsets: still uncovered positive samples $\mathcal{X}^+ \setminus \mathcal{X}_{\mathcal{S}}^+$ each with weight $\omega_+$, already covered positive samples $\mathcal{X}_{\mathcal{S}}^+$ each with weight $-\beta_2$, and negative samples $\mathcal{X}^-$ each with weight $-\beta_0$. As shown in the Appendix, $\omega_+ > 0$ is ensured by choosing $\beta_1 > (e - 1)\beta_2$. Let $\omega_i \in \{\omega_+, -\beta_0, -\beta_2\}$ be weight of the $i$-th sample. Then Equation (5) is equivalent to:

$$
v(\mathcal{R}) = \sum_{i=1}^n \omega_i \mathbb{1}_{\mathcal{R} \subseteq \mathbf{x}_i} - \lambda |\mathcal{R}|
\tag{6}
$$

in which $v(\mathcal{R})$ is short for $v(\mathcal{R}|\mathcal{S}; \alpha)$ and $\mathcal{R} \subseteq \mathbf{x}_i$ is short for $\mathcal{R} \subseteq \{j | x_{i,j} = 1\}$. Intuitively, the goal is to find a short rule to maximize the total weight of samples covered by it. We further rewrite Equation (6) as:

$$
v(\mathcal{R}) = \sum_{i=1}^n \omega_i + \sum_{i:\, \omega_i < 0} -\omega_i \mathbb{1}_{\mathcal{R} \not\subseteq \mathbf{x}_i} - \sum_{i:\, \omega_i > 0} \omega_i \mathbb{1}_{\mathcal{R} \not\subseteq \mathbf{x}_i} - \lambda |\mathcal{R}|
\tag{7}
$$

Define $u(\mathcal{R}) = \sum_{i:\, \omega_i < 0} -\omega_i \mathbb{1}_{\mathcal{R} \not\subseteq \mathbf{x}_i}$ and $w(\mathcal{R}) = \sum_{i:\, \omega_i > 0} \omega_i \mathbb{1}_{\mathcal{R} \not\subseteq \mathbf{x}_i} + \lambda |\mathcal{R}|$. We get the following result.

**Proposition 3.** $u(\mathcal{R})$ *and* $w(\mathcal{R})$ *are non-negative monotone submodular functions.*

*Proof.* Observe that $\{i | \mathcal{R} \not\subseteq \mathbf{x}_i\} = \bigcup_{j \in \mathcal{R}} \{i | \{j\} \not\subseteq \mathbf{x}_i\}$, i.e., the samples **excluded** by $\mathcal{R}$ are the union of samples excluded by each element of $\mathcal{R}$. Then $u(\mathcal{R})$ is a weighted coverage function, which is non-negative monotone submodular. Similarly $w(\mathcal{R})$ is the sum of a weighted coverage function and a non-negative modular function, therefore it is also non-negative monotone submodular. $\quad\square$

Then maximizing $v(\mathcal{R})$ is equivalent to maximizing the difference between submodular functions, $u(\mathcal{R}) - w(\mathcal{R})$. This is not surprising, because any set function can be expressed as a difference of two submodular functions (called a DS function) [37]. However, finding such a decomposition requires exponential complexity for a general set function. This lucky discovery opens the door to nontrivial optimization algorithms based on the submodularity of $u(\mathcal{R})$ and $w(\mathcal{R})$. In this work, we take a minorize-maximization (MM) approach proposed by Iyer and Bilmes [25]. For a submodular function $f : 2^V \to \mathbb{R}_{\geq 0}$, tight lower and upper bound approximations of $f$ at $X \subseteq V$ are given by following modular functions [38, 26].

$$h_{f,X}^\pi(Y) := \sum_{j \in Y} f(S_{\pi^{-1}(j)}^\pi) - f(S_{\pi^{-1}(j)-1}^\pi) \leq f(Y), \forall Y \subseteq V \tag{8}$$

$$m_{f,X}^1(Y) := f(X) - \sum_{j \in X \setminus Y} f(j|X \setminus \{j\}) + \sum_{j \in Y \setminus X} f(j|\emptyset) \geq f(Y), \forall Y \subseteq V \tag{9}$$

$$m_{f,X}^2(Y) := f(X) - \sum_{j \in X \setminus Y} f(j|V \setminus \{j\}) + \sum_{j \in Y \setminus X} f(j|X) \geq f(Y), \forall Y \subseteq V, \tag{10}$$

where $S^\pi$ is a chain obtained by applying a permutation $\pi : [|V|] \to V$ to the ground set $V$ subject to $S_0^\pi = \emptyset$, $S_j^\pi = \{\pi(1), \ldots, \pi(j)\}$ and $S_{|X|}^\pi = X$. With these bounds, the MM idea works by iteratively updating the solution by optimizing (sub)modular surrogates of the DS objective function. In our case, $u(\mathcal{R}) - w(\mathcal{R})$ is approximately maximized using the modular-modular procedure [25] detailed in Algorithm 2. At each iteration, lower bounding surrogates of $u(\mathcal{R}) - w(\mathcal{R})$ are obtained via replacing $u(\mathcal{R})$ by its modular lower bound and replacing $w(\mathcal{R})$ by its modular upper bounds. These modular surrogates are maximized exactly through selecting all positive elements.

---

**Algorithm 2** DS-OPT$(\mathcal{R}, u, w)$

---

1   **while true do**
2     $\mathcal{R}' \leftarrow \mathcal{R}$
3     Choose a permutation $\pi$ of $[d]$ to form the chain $S^\pi$
4     For $\forall j \in [d]$, compute $h_{u,\mathcal{R}}^\pi(j) = u(S_{\pi^{-1}(j)}^\pi) - u(S_{\pi^{-1}(j)-1}^\pi)$
5     For $\forall j \in \mathcal{R}$, compute $m_{w,\mathcal{R}}^1(j) = w(j|\mathcal{R} \setminus \{j\})$ and $m_{w,\mathcal{R}}^2(j) = w(j|[d] \setminus \{j\})$
6     For $\forall j \notin \mathcal{R}$, compute $m_{w,\mathcal{R}}^1(j) = w(j|\emptyset)$ and $m_{w,\mathcal{R}}^2(j) = w(j|\mathcal{R})$
7     $\mathcal{R}_1 \leftarrow \{j | h_{u,\mathcal{R}}^\pi(j) - m_{w,\mathcal{R}}^1(j) > 0\}$, $\mathcal{R}_2 \leftarrow \{j | h_{u,\mathcal{R}}^\pi(j) - m_{w,\mathcal{R}}^2(j) > 0\}$
8     $\mathcal{R} \leftarrow \arg\max_{R \in \{\mathcal{R}_1, \mathcal{R}_2\}} u(R) - w(R)$
9     **if** $\mathcal{R} = \mathcal{R}'$ **then break end if**
10  **end while**
11  **Output:** $\mathcal{R}$

---

To further reduce the possibility of stucking in local maxima, the MM procedure is augmented with a restricted exact search step and a swap-based local search heuristic as sketched in Algorithm 3. At line 6-7, the current solution is enlarged into an active set of size $M$ and an exhaustive subset search restricted on this active set is conducted. The aim of this exact search step is to produce good initialization for DS-OPT, and we find empirically that the gain ratio $u(j|\mathcal{R})/w(j|\mathcal{R})$ is a good criterion for including new features in the active set. In our implementation, we set $M = 16$ and carry out the exact search with customized branch-and-bound (BnB). At line 9, the SwapLocalSearch subprocedure tries to improve the solution by adding, removing or replacing a feature.

## 5   Experiments

We evaluate the proposed approach in terms of predictive performance, interpretability, scalability, and approximation quality.

### 5.1   Setup

**Datasets.**   Our experimental study is conducted on 20 public datasets. Fifteen of them are from the UCI repository [16], and the other five are variants of the ProPublica recidivism dataset (COMPAS)

---

**Algorithm 3** Local combinatorial search

---

  1  **Input:** Training data $\{(\mathbf{x}_i, y_i)\}_{i=1}^n$, objective function $v(\mathcal{R})$, threshold $M$ for exact search
  2  Decompose $v(\mathcal{R}) \equiv u(\mathcal{R}) - w(\mathcal{R})$
  3  Initialize $\mathcal{R} \leftarrow \emptyset$
  4  **while true do**
  5     $\tilde{\mathcal{R}} \leftarrow \mathcal{R} \,;\; \mathcal{R}' \leftarrow \mathcal{R}$
  6     **if** $|\tilde{\mathcal{R}}| < M$ **then** $\tilde{\mathcal{R}} \leftarrow \text{Enlarge}(\tilde{\mathcal{R}}, M, u, w)$ **end if**
  7     **if** $|\tilde{\mathcal{R}}| \leq M$ **then** $\mathcal{R} \leftarrow \text{BestSubset}(\tilde{\mathcal{R}}, u, w)$ **end if**
  8     $\mathcal{R} \leftarrow \text{DS-OPT}(\mathcal{R}, u, w)$
  9     $\mathcal{R} \leftarrow \text{SwapLocalSearch}(\mathcal{R}, u, w)$
10     **if** $\mathcal{R} = \mathcal{R}'$ **then break end if**
11  **end while**
12  **Output:** $\mathcal{R}$

---

[29] and the Fair Isaac credit risk dataset (FICO) [21] that have been used in recent work to evaluate interpretable rule-based models [34]. We binarize the features of all datasets in exactly the same way as [14]. Specifically, for categorical features, two features $x_j = z$ and $x_j \neq z$ are created for each category $z$. For numerical features, we create comparison features $x_j \leq z$ and $x_j > z$ via choosing sample deciles as the boundary value $z$. The number of binary features of each dataset after preprocessing is given in Table 1, which is up to 2922. Note that the negative and comparison features produced in this preprocessing method will make the feature matrices dense and make the features highly correlated, which poses major challenges to rule set learning algorithms.

**Baselines.** The proposed method is compared with recently developed algorithms that explicitly optimize the interpretability of rule sets[1], including CG [14] and BRS [49], as well as a classical sequential cover algorithm, RIPPER [13]. The official light version of CG implementation [5] is used in our experiments, which solves the rule generation subproblem using heuristic beam search instead of IP solver. This is exactly comparable to our method because we also rely on approximate algorithm, which is more practical than calling a solver. For BRS, we use a third-party implementation [32] that supports generating candidate rules based on random forest instead of frequent itemset miner, because we find that the latter has poor scalability. For RIPPER, an open source Python implementation [36] is chosen. In addition, we also include the widely used CART and random forest (RF) algorithms implemented in the scikit-learn package [40] to illustrate the competitiveness of rule set models.

**Parameter tuning.** We estimate numerical results based on 10-fold stratified cross-validation (CV). In each CV fold, we use grid search to optimize the hyperparameters of each algorithm on the training split. For the method proposed in this paper, we fix $\beta_0 = \beta_1 = 1$ and optimize the remaining hyperparameters $\beta_2 \in \{0.5, 0.1, 0.01\}$, $\lambda \in \{0.1, 1, 4, 8, 16, 64\}$ and $K \in \{8, 16, 32\}$. The hyperparameters of CG include the strength of complexity penalty and the beam width, for which we sweep in $\{0.001, 0.002, 0.005\}$ and $\{10, 20\}$, respectively. For RIPPER, the proportion of training set used for pruning is varied in $\{0.2, 0.25, \ldots, 0.6\}$. For BRS, the maximum length of a rule is chosen from $\{3, 5\}$. For CART and RF, we tune the minimum number of samples at leaf nodes from 1 to 100 and fix the number of trees in RF to be 100.

### 5.2 Numerical results

**Predictive performance.** The predictive performance measured by average test accuracy over 10 folds is reported in Table 2. BRS is not available on magic and gas because it ran beyond our time limit, and CG is not available on COMPAS because we encountered an LP solver error. On small datasets, the accuracy of our submodular optimization based approach matches the CG based approach, and both of them are top-performing rule set learners that are as accurate as the tree-based model, CART. The differences should not be overly concerned here because the variances are high due to small sample sizes. Notably, our method achieves 100% test accuracy on tic-tac-toe and mushroom, for which well-known perfect rule set solutions exist. On larger datasets, we observe that

---

[1]We failed to run IDS [28] and DRS [51] on most of the datasets.

Table 1: Predictive performance measured by average test accuracy (%).

| Dataset | #samples | #features | Ours | RIPPER | BRS | CG | CART | RF |
|---|---|---|---|---|---|---|---|---|
| tic-tac-toe | 958 | 54 | $100.0_{(0.0)}$ | $99.7_{(0.7)}$ | $100.0_{(0.0)}$ | $100.0_{(0.0)}$ | $94.2_{(1.9)}$ | $99.1_{(0.9)}$ |
| liver | 345 | 104 | $69.5_{(5.1)}$ | $66.0_{(5.8)}$ | $60.6_{(8.3)}$ | $68.7_{(5.4)}$ | $68.6_{(6.3)}$ | $73.9_{(9.3)}$ |
| heart | 303 | 118 | $82.2_{(7.7)}$ | $76.2_{(7.7)}$ | $79.7_{(7.5)}$ | $78.0_{(6.8)}$ | $82.2_{(6.1)}$ | $82.8_{(7.1)}$ |
| ionosphere | 351 | 566 | $91.4_{(5.4)}$ | $87.2_{(7.5)}$ | $85.0_{(4.2)}$ | $90.6_{(4.4)}$ | $89.5_{(3.3)}$ | $94.0_{(3.4)}$ |
| ILPD | 583 | 160 | $71.4_{(0.8)}$ | $57.8_{(7.7)}$ | $69.0_{(5.3)}$ | $71.7_{(3.4)}$ | $69.4_{(6.4)}$ | $71.2_{(4.0)}$ |
| WDBC | 569 | 540 | $94.0_{(4.8)}$ | $94.7_{(1.6)}$ | $93.9_{(1.2)}$ | $94.7_{(3.4)}$ | $93.5_{(3.8)}$ | $97.0_{(3.6)}$ |
| pima | 768 | 134 | $75.4_{(4.3)}$ | $75.9_{(3.3)}$ | $72.2_{(3.3)}$ | $74.0_{(3.4)}$ | $75.4_{(5.5)}$ | $76.9_{(3.3)}$ |
| transfusion | 748 | 64 | $78.1_{(3.2)}$ | $78.2_{(2.7)}$ | $77.1_{(5.1)}$ | $78.2_{(3.6)}$ | $78.7_{(2.8)}$ | $79.7_{(2.8)}$ |
| banknote | 1372 | 72 | $98.7_{(1.0)}$ | $92.8_{(2.4)}$ | $91.1_{(2.5)}$ | $98.8_{(0.9)}$ | $99.1_{(1.2)}$ | $99.6_{(0.6)}$ |
| mushroom | 8124 | 224 | $100.0_{(0.0)}$ | $100.0_{(0.0)}$ | $99.7_{(0.2)}$ | $99.9_{(0.1)}$ | $100.0_{(0.0)}$ | $100.0_{(0.0)}$ |
| COMPAS-2016 | 5020 | 30 | $66.5_{(2.3)}$ | $57.7_{(1.0)}$ | $63.4_{(1.7)}$ | $66.7_{(2.2)}$ | $66.2_{(2.2)}$ | $66.6_{(2.5)}$ |
| COMPAS-binary | 6907 | 24 | $67.0_{(1.5)}$ | $56.0_{(0.6)}$ | $65.5_{(1.7)}$ | $66.4_{(1.9)}$ | $67.3_{(1.5)}$ | $67.3_{(1.6)}$ |
| FICO-binary | 10459 | 34 | $71.2_{(1.1)}$ | $60.1_{(1.2)}$ | $70.5_{(1.1)}$ | $71.1_{(1.2)}$ | $71.9_{(1.4)}$ | $72.3_{(1.4)}$ |
| COMPAS | 12381 | 180 | $\mathbf{73.3}_{(1.3)}$ | $72.3_{(1.5)}$ | $70.7_{(1.1)}$ | N/A | $72.2_{(1.4)}$ | $73.8_{(1.1)}$ |
| FICO | 10459 | 312 | $70.4_{(1.2)}$ | $69.1_{(1.9)}$ | $70.1_{(0.9)}$ | $\mathbf{71.0}_{(0.7)}$ | $70.9_{(1.1)}$ | $72.3_{(0.8)}$ |
| adult | 48842 | 262 | $\mathbf{84.4}_{(0.6)}$ | $83.3_{(0.9)}$ | $80.3_{(1.4)}$ | $82.8_{(0.4)}$ | $83.7_{(0.4)}$ | $84.7_{(0.5)}$ |
| bank-market | 11162 | 174 | $\mathbf{84.4}_{(0.8)}$ | $82.9_{(1.1)}$ | $76.9_{(1.2)}$ | $82.3_{(0.9)}$ | $83.0_{(1.0)}$ | $85.2_{(0.9)}$ |
| magic | 19020 | 180 | $\mathbf{84.6}_{(0.8)}$ | $82.2_{(1.3)}$ | N/A | $80.8_{(1.0)}$ | $84.7_{(0.5)}$ | $86.7_{(0.5)}$ |
| musk | 6598 | 2922 | $\mathbf{97.3}_{(0.8)}$ | $96.1_{(0.8)}$ | $90.2_{(2.0)}$ | $95.0_{(0.7)}$ | $96.0_{(0.9)}$ | $97.7_{(0.6)}$ |
| gas | 13910 | 2304 | $98.2_{(0.4)}$ | $\mathbf{99.0}_{(0.4)}$ | N/A | $95.9_{(0.7)}$ | $99.0_{(0.3)}$ | $99.8_{(0.1)}$ |

our method generally demonstrates superiority over other rule set learners. Overall, the accuracy gaps between our method and the uninterpretable RF are within 3% on all datasets except liver.

Table 2: Interpretability measured by number of rules, number of literals, and overlap among rules.

| Dataset | #Rules | | | | #Literals | | | | Overlap (%) | | |
|---|---|---|---|---|---|---|---|---|---|---|---|
| | Ours | RIPPER | CG | CART | Ours | RIPPER | CG | CART | Ours | RIPPER | CG |
| tic-tac-toe | 8.0 (0.0) | 9.5 (1.4) | 8.0 (0.0) | 69.9 (3.6) | 24.0 (0.0) | 31.1 (5.8) | 24.3 (0.5) | 138.8 (7.1) | 2.3 (1.2) | 52.8 (8.1) | 23.3 (0.5) |
| liver | 18.0 (2.4) | 2.1 (0.7) | 14.5 (1.2) | 5.0 (0.0) | 83.8 (10.5) | 7.1 (3.3) | 58.5 (4.9) | 9.0 (0.0) | 7.5 (4.9) | 28.0 (17.7) | 9.7 (1.7) |
| heart | 2.1 (0.3) | 4.0 (1.1) | 10.3 (0.8) | 11.4 (1.1) | 4.4 (1.3) | 11.0 (3.8) | 41.5 (3.2) | 21.8 (2.1) | 16.8 (7.7) | 48.4 (4.9) | 27.4 (2.4) |
| ionosphere | 2.0 (0.7) | 3.6 (0.8) | 4.3 (0.8) | 24.7 (2.1) | 8.0 (2.4) | 12.5 (3.1) | 20.3 (3.8) | 48.4 (4.2) | 3.4 (5.0) | 57.2 (7.9) | 32.1 (7.1) |
| ILPD | 1.1 (0.3) | 2.6 (0.5) | 2.0 (0.0) | 4.3 (0.5) | 0.2 (0.6) | 7.0 (1.5) | 3.0 (0.0) | 7.6 (1.0) | 0.0 (0.0) | 31.7 (6.7) | 0.0 (0.1) |
| WDBC | 8.0 (1.1) | 5.0 (1.1) | 5.3 (0.6) | 7.9 (1.0) | 27.7 (3.0) | 10.6 (3.0) | 13.4 (1.6) | 14.8 (2.0) | 2.4 (6.0) | 35.0 (5.3) | 26.8 (1.0) |
| pima | 3.2 (2.2) | 3.6 (1.3) | 6.7 (1.7) | 10.1 (0.6) | 10.0 (10.7) | 13.0 (6.5) | 20.0 (6.5) | 19.2 (1.1) | 2.6 (3.1) | 27.0 (8.2) | 5.4 (1.2) |
| transfusion | 1.4 (0.8) | 2.1 (0.7) | 2.7 (0.5) | 11.0 (0.5) | 4.3 (2.8) | 9.0 (3.0) | 7.9 (1.6) | 21.0 (0.9) | 0.0 (0.0) | 21.3 (13.1) | 0.8 (0.5) |
| banknote | 8.7 (1.3) | 7.4 (1.3) | 4.0 (0.0) | 28.2 (0.7) | 32.8 (5.8) | 21.0 (4.2) | 10.9 (1.4) | 55.4 (5.4) | 0.1 (0.3) | 41.0 (3.2) | 9.5 (0.7) |
| mushroom | 3.9 (0.3) | 6.1 (1.1) | 5.0 (0.0) | 14.1 (0.6) | 8.4 (1.3) | 10.9 (1.1) | 7.0 (0.0) | 27.2 (1.1) | 0.0 (0.0) | 41.6 (15.8) | 21.0 (0.2) |
| COMPAS-2016 | 11.8 (4.6) | 9.0 (1.6) | 3.0 (0.0) | 31.0 (0.7) | 41.9 (20.1) | 31.4 (7.0) | 6.0 (0.0) | 61.0 (1.3) | 0.0 (0.1) | 30.9 (0.2) | 0.5 (0.2) |
| COMPAS-binary | 11.4 (1.3) | 12.0 (1.7) | 2.9 (0.3) | 78.0 (1.1) | 42.2 (6.3) | 48.1 (8.8) | 5.4 (0.9) | 155.0 (2.1) | 0.1 (0.2) | 32.0 (0.7) | 0.0 (0.0) |
| FICO-binary | 21.6 (3.3) | 16.7 (2.1) | 2.0 (0.0) | 158.5 (2.7) | 134.0 (23.8) | 106.8 (14.3) | 4.0 (0.0) | 316.0 (5.4) | 0.3 (0.3) | 37.8 (0.9) | 0.0 (0.0) |
| COMPAS | 5.5 (2.7) | 12.0 (2.7) | N/A | 85.9 (2.3) | 24.0 (15.8) | 66.0 (15.3) | N/A | 170.8 (4.7) | 0.2 (0.3) | 21.2 (3.5) | N/A |
| FICO | 16.0 (5.5) | 16.7 (3.9) | 1.1 (0.3) | 69.5 (1.4) | 118.6 (39.3) | 99.7 (26.6) | 1.4 (1.2) | 138.0 (2.7) | 3.5 (1.3) | 39.8 (3.8) | 0.0 (0.0) |
| adult | 9.1 (3.1) | 42.7 (15.2) | 2.0 (0.0) | 398.3 (4.9) | 83.4 (30.7) | 337.0 (128.9) | 5.7 (0.5) | 795.6 (9.9) | 0.8 (0.6) | 25.8 (7.0) | 1.8 (0.4) |
| bank-market | 17.6 (3.2) | 43.8 (7.1) | 11.0 (0.0) | 289.0 (2.8) | 118.5 (15.1) | 269.1 (49.2) | 18.7 (0.8) | 577.0 (5.7) | 3.6 (1.9) | 43.8 (3.6) | 7.8 (0.4) |
| magic | 19.1 (6.6) | 52.3 (10.6) | 2.7 (0.5) | 398.9 (4.7) | 136.1 (49.2) | 391.3 (75.0) | 6.2 (0.4) | 796.8 (9.3) | 2.8 (1.7) | 50.0 (0.0) | 7.6 (2.8) |
| musk | 8.9 (1.8) | 19.6 (1.5) | 5.0 (0.4) | 180.0 (7.1) | 61.2 (14.8) | 101.4 (7.6) | 21.0 (1.9) | 359.0 (14.2) | 0.8 (0.5) | 36.9 (3.9) | 2.7 (1.4) |
| gas | 13.1 (1.0) | 24.2 (1.8) | 4.0 (0.0) | 172.2 (2.3) | 74.2 (4.2) | 106.8 (11.6) | 14.7 (1.5) | 343.4 (4.7) | 15.2 (1.7) | 50.0 (0.0) | 22.1 (3.0) |

**Interpretability.** The interpretability of learned models is compared in terms of three metrics: the number of rules, the total number of literals in all rules, and the overlap among rules. In our notation, the former two complexity metrics are $|\mathcal{S}|$ and $\sum_{\mathcal{R} \in \mathcal{S}} |\mathcal{R}|$, respectively. The overlap is measured as the fraction of samples covered by more than one of the rules, i.e., $|\{i|\tilde{P}_{\mathcal{S}}(\mathbf{x}_i) > 1\}|/n$. For CARTs, we treat their root-to-leaf paths as rules. The complexities of RFs are generally high because it consists of many trees, therefore we do not report them here. Note that the overlap among rules generated from a CART is zero because these rules form a partition of the feature space. We leave BRS out here as its predictive performance on larger datasets is incomparable to remaining methods.

The results averaged over 10 folds are reported in Table 2. We observe that CG usually produced rule sets with lowest complexity, while our method is competitive when compared to RIPPER. Our method learned notably simpler rule sets than RIPPER on large datasets except FICO. When it comes to the overlap among learned rules, our method dominates the competition because it explicitly penalizes the overlap in the objective function. We notice that RIPPER is particularly problematic if overlap is the concern, as $> 20\%$ samples are covered more than once on all datasets. The learned 100%-accurate rule sets on the mushroom and tic-tac-toe datasets are illustrated in Table 3. Remarkably, the rule set for mushroom is even simpler than the ground truth provided in the dataset description file [1], as the third rule consists of only two conditions instead of three.

**Scalability.** Our method scales linearly with the dimensionality, the sample size and the number of iterations. In our experience, the local search algorithm for the subproblem usually terminates at a stationary point in a few iterations. Bit vectors are used in our implementation to process a large number of samples efficiently. Therefore for our algorithm the dimensionality is the main bottleneck in scalability. Figure 1 shows the training time of our method on the musk and gas datasets when using randomly sampled subsets of features. As the number of features grows, the training time demonstrates a roughly linear trend as expected. Our method is thus more practical when compared to solving the subproblem with exact algorithms, e.g., branch-and-bound, because it is well known that exact algorithms for general combinatorial optimization problems scale poorly with dimensionality.

The average running time of each method for fitting each dataset under typical hyperparameters found in cross-validation is reported in the Appendix. In summary, our method is competitive with other rule set learners but is slower than well-optimized tree-based methods. Further boosting of scalability may be achieved through drawing inspiration from scalable submodular optimization techniques [27], and we leave this for future work.

Table 3: Examples of learned rule sets.

| mushroom |
| --- |
| odor != a AND odor != l AND odor != n |
| spore_print_color = r |
| gill_size = n AND stalk_surface_below_ring = y |
| cap_color = w AND population = c |

| tic-tac-toe |
| --- |
| top-right = x AND middle-middle = x AND bottom-left = x |
| top-left = x AND middle-middle = x AND bottom-right = x |
| middle-left = x AND middle-middle = x AND middle-right = x |
| bottom-left = x AND bottom-middle = x AND bottom-right = x |
| top-left = x AND top-middle = x AND top-right = x |
| top-left = x AND middle-left = x AND bottom-left = x |
| top-right = x AND middle-right = x AND bottom-right = x |
| top-middle = x AND middle-middle = x AND bottom-middle = x |

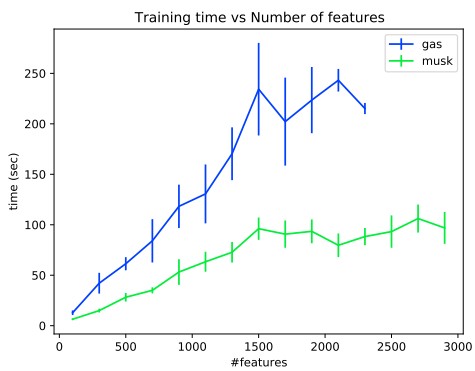

Figure 1: Scalability test.

**Approximation quality.** To examine the effectiveness of Algorithm 3, we conduct a comparison between approximate subproblem solving and exact subproblem solving under the same typical hyperparameter settings. On nine low-dimensional datasets, all of the subproblems could be exactly solved within 10 minutes using our customized BnB procedure. As the subproblems for remaining higher-dimensional datasets could not be solved to optimality with practical running time, a time limit of 10 minutes is specified for the BnB subproblem solver. For each dataset, we record the values of objective function $V(\mathcal{S})$ achieved by learned rule sets under the two setups and calculate the relative gaps between solutions as $[V(\mathcal{S}_{\text{bnb}}) - V(\mathcal{S}_{\text{approx}})]/V(\mathcal{S}_{\text{bnb}})$. Results are summarized in the Appendix.

The relative gaps are small in general and do not grow with the dimensionality, indicating that the proposed approximate algorithm works well on these datasets. For instance, the two setups produced exactly the same rule sets on five of the low-dimensional datasets. Interestingly, it turns out that approximate subproblem solving is not necessarily worse than the exact counterpart. This is because the outer loop (Algorithm 1) is also an approximation procedure, which does not guarantee that better subproblem solutions will help build a better solution to the main problem.

## 6    Limitations

**Approximation guarantee.**    The main limitation of this work is the invalidity of approximation guarantee (Proposition 2) under inexact subproblem solving. Nonetheless, the approximation guarantee is still relevant in the case that the optimal solution to the subproblem can be found. This is possible if we restrict the space of rules to a tractable extent. For example, if we move back to the pre-mining approach in which rules can only be picked from a given pool (e.g., rules extracted from a random forest), the guarantee will still hold for this modified problem. Furthermore, the previous discussion of approximation quality shows that empirically such invalidity will not necessarily lead to much loss of performance. We are planning to analyze this phenomenon theoretically in future work.

**Multiclass extension.**    Another more subtle limitation is the extension to multiclass classification. We have suggested the model-agnostic one-versus-one (OvO) transformation, which only requires the base classifiers to produce a binary class label, rather than a real-valued confidence score required by the one-versus-rest (OvR) approach. However, both OvR and OvO may suffer from ambiguities and class imbalance. Moreover, the OvO reduction is not very efficient because $K(K-1)/2$ base classifiers must be trained for a $K$-class classification task. We are investigating if there is a more scalable approach to extend this work to multiclass problems.

**Societal impact.**    On the positive side, the underlying decision logic of an interpretable model such as a rule set is more transparent for humans to understand. When used as a data exploration tool, such transparency enables people to check easily if the fitted model has discovered some new knowledge about the data. When deployed to make automatic decisions in high-stake scenarios, such transparency helps people to diagnose if there are potential biases or risks in the model and to explain the reason behind a specific decision. On the negative side, a rule set learned by optimizing accuracy and interpretability will not automatically attain other properties requested by trustworthy AI, such as fairness and robustness. In addition, a learned rule "IF X=x0 AND Y=y1 THEN Z=1" may not conform to the causal mechanism underlying the variables X, Y and Z. This will lead to misunderstandings if the rules are not interpreted properly.

## 7    Conclusions

In this paper, we propose a new perspective on interpretable rule set learning based on submodular optimization. The learning task is decomposed into two related subset selection problems, where the main problem selects a subset from all possible rules to optimize a regularized submodular objective, and the subproblem selects a feature subset to form a rule. The optimization criteria of the subproblem is derived based on a distorted greedy algorithm for maximizing the main objective, and we observe that this criteria can be expressed as the difference between two submodular functions. Based on this finding, an iterative local combinatorial search algorithm is designed to solve the subproblem approximately. The effectiveness of our method is evaluated through a comprehensive experimental study on 20 datasets. Directions for future work include warm-starting the search with rules generated by random forests and extending the method to rule-based regression tasks.

## Acknowledgments

We thank the anonymous reviewers at NeurIPS 2021 for their insightful feedback.

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
