# A  Appendix

## A.1  Other rule set learning objectives

We show that several other learning objectives for rule sets may reduce to (3).

### A.1.1  0-1 loss with complexity penalty

Let $l(\hat{y}, y) = \mathbb{I}(\hat{y} \neq y)$ be the 0-1 loss and $\Omega(\mathcal{S}) = \lambda \sum_{\mathcal{R} \in \mathcal{S}} |\mathcal{R}|$ be the complexity penalty. Then

$$\sum_{i=1}^{n} l\left(P_{\mathcal{S}}(\mathbf{x}_i), y_i\right) + \Omega(\mathcal{S})$$

$$= \left|\mathcal{X}^+ \setminus \mathcal{X}_{\mathcal{S}}^+\right| + \left|\mathcal{X}_{\mathcal{S}}^-\right| + \lambda \sum_{\mathcal{R} \in \mathcal{S}} |\mathcal{R}|$$

$$\leq \left|\mathcal{X}^+ \setminus \mathcal{X}_{\mathcal{S}}^+\right| + \sum_{\mathcal{R} \in \mathcal{S}} \left|\mathcal{X}_{\{\mathcal{R}\}}^-\right| + \lambda \sum_{\mathcal{R} \in \mathcal{S}} |\mathcal{R}| =: L_1(\mathcal{S}).$$

It is obvious that $L_1(\mathcal{S})$ is equal to $L(\mathcal{S})$ with $\beta_0 = \beta_1 = 1$ and $\beta_2 = 0$. Therefore, minimizing $L(\mathcal{S})$ can be interpreted as minimizing an upper bounding surrogate of the penalized 0-1 loss.

### A.1.2  0-1 loss with overlap penalty

Let $\Omega(\mathcal{S}) = \eta \left( \sum_{\mathcal{R} \in \mathcal{S}} |\mathcal{X}_{\{\mathcal{R}\}}| - |\mathcal{X}_{\mathcal{S}}| \right)$ be the overlap penalty, in which $\mathcal{X}_{\mathcal{S}} := \{i | P_{\mathcal{S}}(\mathbf{x}_i)\}$. If $\eta \leq 1$, we have

$$\sum_{i=1}^{n} l\left(P_{\mathcal{S}}(\mathbf{x}_i), y_i\right) + \Omega(\mathcal{S})$$

$$= \left|\mathcal{X}^+ \setminus \mathcal{X}_{\mathcal{S}}^+\right| + \left|\mathcal{X}_{\mathcal{S}}^-\right| + \eta \left( \sum_{\mathcal{R} \in \mathcal{S}} |\mathcal{X}_{\{\mathcal{R}\}}| - |\mathcal{X}_{\mathcal{S}}| \right)$$

$$= \left|\mathcal{X}^+\right| - \left|\mathcal{X}_{\mathcal{S}}^+\right| + \left|\mathcal{X}_{\mathcal{S}}^-\right| - \eta(|\mathcal{X}_{\mathcal{S}}^+| + |\mathcal{X}_{\mathcal{S}}^-|) + \eta \sum_{\mathcal{R} \in \mathcal{S}} \left|\mathcal{X}_{\{\mathcal{R}\}}^+\right| + \left|\mathcal{X}_{\{\mathcal{R}\}}^-\right|$$

$$= \left|\mathcal{X}^+\right| - (1 + \eta)\left|\mathcal{X}_{\mathcal{S}}^+\right| + (1 - \eta)\left|\mathcal{X}_{\mathcal{S}}^-\right| + \eta \sum_{\mathcal{R} \in \mathcal{S}} \left|\mathcal{X}_{\{\mathcal{R}\}}^+\right| + \left|\mathcal{X}_{\{\mathcal{R}\}}^-\right|$$

$$\leq \left|\mathcal{X}^+\right| - (1 + \eta)\left|\mathcal{X}_{\mathcal{S}}^+\right| + (1 - \eta) \sum_{\mathcal{R} \in \mathcal{S}} \left|\mathcal{X}_{\{\mathcal{R}\}}^-\right| + \eta \sum_{\mathcal{R} \in \mathcal{S}} \left|\mathcal{X}_{\{\mathcal{R}\}}^+\right| + \left|\mathcal{X}_{\{\mathcal{R}\}}^-\right|$$

$$= \left|\mathcal{X}^+\right| - (1 + \eta)\left|\mathcal{X}_{\mathcal{S}}^+\right| + \sum_{\mathcal{R} \in \mathcal{S}} \eta \left|\mathcal{X}_{\{\mathcal{R}\}}^+\right| + \left|\mathcal{X}_{\{\mathcal{R}\}}^-\right| =: L_2(\mathcal{S}).$$

It can be observed that $L_2(\mathcal{S})$ is equal to $L(\mathcal{S})$ with $\beta_0 = \beta_1 = 1$, $\beta_2 = \eta$ and $\lambda = 0$.

### A.1.3  Hamming loss

The Hamming loss employed by Dash et al. [14] is equal to $L(\mathcal{S})$ with $\beta_0 = \beta_1 = 1$, $\beta_2 = 0$ and $\lambda = 0$.

## A.2 Guidance for hyperparameter tuning

As pointed out in Section 4.2, the multiplier $\alpha := (1 - 1/K)^{K-k} \geq 1/e$ for $K \in \mathbb{N}^+$ and $k = 1, \ldots, K$. Then $\omega := \alpha(\beta_1 + \beta_2) - \beta_2 > 0$ is ensured by choosing $\beta_1 > (e-1)\beta_2$, as:

$$
\begin{aligned}
& \alpha(\beta_1 + \beta_2) - \beta_2 \\
\geq & \frac{1}{e}(\beta_1 + \beta_2) - \beta_2 \\
> & \frac{1}{e}\left[(e-1)\beta_2 + \beta_2\right] - \beta_2 = 0.
\end{aligned}
$$

## A.3 Additional algorithms

The output of Algorithm 1 is refined by the following local search algorithm, which tries to improve the objective $V(\mathcal{S})$ through adding, removing, or replacing a rule.

---

**Algorithm 4** Refine a rule set

---

1 **Input:** Training data $\{(\mathbf{x}_i, y_i)\}_{i=1}^n$, hyperparameters $(\boldsymbol{\beta}, \lambda)$, cardinality $K$, initial solution $\mathcal{S}$
2 **while true do**
3    $\mathcal{S}' \leftarrow \mathcal{S}$
4    **for** $k = |\mathcal{S}|$ **to** $K-1$ **do**
5       Define $v(\mathcal{R}) = g(\mathcal{R}|\mathcal{S}) - c(\mathcal{R})$
6       Solve $\mathcal{R}^\star \leftarrow \arg\max_{\mathcal{R} \subseteq [d]} v(\mathcal{R})$
7       **if** $v(\mathcal{R}^\star) > 0$ **then** $\mathcal{S} \leftarrow \mathcal{S} \cup \{\mathcal{R}^\star\}$ **end if**
8    **end for**
9    **for** $\mathcal{R}' \in \mathcal{S}$ **do**
10      $\mathcal{S} \leftarrow \mathcal{S} \setminus \{\mathcal{R}'\}$
11      Define $v(\mathcal{R}) = g(\mathcal{R}|\mathcal{S}) - c(\mathcal{R})$
12      Solve $\mathcal{R}^\star \leftarrow \arg\max_{\mathcal{R} \subseteq [d]} v(\mathcal{R})$
13      **if** $v(\mathcal{R}^\star) > 0$ **then** $\mathcal{S} \leftarrow \mathcal{S} \cup \{\mathcal{R}^\star\}$ **end if**
14    **end for**
15    **if** $\mathcal{S} = \mathcal{S}'$ **then break end if**
16 **end while**
17 **Output:** $\mathcal{S}$

---

The Enlarge subprocedure in Algorithm 3 expands the current solution into an active set of size $M$.

---

**Algorithm 5** Enlarge($\tilde{\mathcal{R}}, M, u, w$)

---

1 **for** $k = |\tilde{\mathcal{R}}|$ **to** $M-1$ **do**
2    $j^\star \leftarrow \arg\max_j u(j|\tilde{\mathcal{R}})/w(j|\tilde{\mathcal{R}})$
3    $\tilde{\mathcal{R}} \leftarrow \tilde{\mathcal{R}} \cup \{j^\star\}$
4 **end for**
5 **Output:** $\tilde{\mathcal{R}}$

---

The SwapLocalSearch subprocedure in Algorithm 3 tries to improve the output of DS-OPT through adding, removing or replacing a feature.

Table 4: Characteristics of datasets used in our experimental study.

| Dataset | #samples | #features | #binarized | #positives | #negatives |
|---|---|---|---|---|---|
| tic-tac-toe | 958 | 9 | 54 | 626 | 332 |
| liver | 345 | 6 | 104 | 145 | 200 |
| heart | 303 | 13 | 118 | 165 | 138 |
| ionosphere | 351 | 34 | 566 | 225 | 126 |
| ILPD | 583 | 10 | 160 | 416 | 167 |
| WDBC | 569 | 30 | 540 | 212 | 357 |
| pima | 768 | 8 | 134 | 268 | 500 |
| transfusion | 748 | 4 | 64 | 178 | 570 |
| banknote | 1372 | 4 | 72 | 610 | 762 |
| mushroom | 8124 | 22 | 224 | 3916 | 4208 |
| COMPAS-2016 | 5020 | 6 | 30 | 2246 | 2774 |
| COMPAS-binary | 6907 | 12 | 24 | 3196 | 3711 |
| FICO-binary | 10459 | 17 | 34 | 5000 | 5459 |
| COMPAS | 12381 | 22 | 180 | 3855 | 8526 |
| FICO | 10459 | 23 | 312 | 5000 | 5459 |
| adult | 48842 | 14 | 262 | 11687 | 37155 |
| bank-market | 11162 | 16 | 174 | 5289 | 5873 |
| magic | 19020 | 10 | 180 | 12332 | 6688 |
| musk | 6598 | 166 | 2922 | 1017 | 5581 |
| gas | 13910 | 128 | 2304 | 6778 | 7132 |

---

**Algorithm 6** SwapLocalSearch$(\mathcal{R}, u, w)$

---

1 **while true do**
2    $\mathcal{R}'' \leftarrow \mathcal{R}$
3    **while** $\exists j \in [d] \setminus \mathcal{R}$ s.t. $v(j|\mathcal{R}) > 0$ **do** $\mathcal{R} \leftarrow \mathcal{R} \cup \{j\}$ **end while**
4    **while** $\exists j \in \mathcal{R}$ s.t. $v(j|\mathcal{R} \setminus \{j\}) \leq 0$ **do** $\mathcal{R} \leftarrow \mathcal{R} \setminus \{j\}$ **end while**
5    **while** $\exists a \in \mathcal{R}, b \in [d] \setminus \mathcal{R}$ s.t. $v(b|\mathcal{R} \setminus \{a\}) > 0$ **do** $\mathcal{R} \leftarrow (\mathcal{R} \setminus \{a\}) \cup \{b\}$ **end while**
6    **if** $\mathcal{R} = \mathcal{R}''$ **then break end if**
7 **end while**
8 **Output:** $\mathcal{R}$

---

### A.4 Dataset details

Table 4 shows several extra dataset characteristics, including the number of original features and the number of positive/negative samples in each dataset.

### A.5 Running time

The average running time in seconds of each method was measured on a MacBook Pro (2019 edition, Intel Core i5). The hyperparameters of the methods were set to their typical values found in cross-validation. Results are summarized in Table 5.

The readers should be aware that the running time comparison based on wall-clock time here is not totally fair, as the implementations of these methods are not optimized to a matching level. Among the baselines, CG, BRS and RIPPER are implemented in Python (with numerical computation offloaded to numpy), while CART and RF are based on a highly optimized Cython backend. Our method is implemented in Golang, a programming language that in general is faster than Python and is slower than C/C++. Nevertheless, this table, jointly with the reported experimental data on accuracy and interpretability, enables us to roughly understand the trade-offs of these methods.

Table 5: Average running time in seconds.

| Dataset | Ours | CG | RIPPER | BRS | CART | RF |
|---|---|---|---|---|---|---|
| tic-tac-toe | 0.794 | 12.815 | 0.204 | 14.833 | 0.002 | 0.097 |
| liver | 4.113 | 62.482 | 0.232 | 19.513 | 0.002 | 0.083 |
| heart | 0.853 | 62.840 | 0.227 | 14.858 | 0.001 | 0.077 |
| ionosphere | 6.064 | 51.475 | 0.914 | 17.304 | 0.008 | 0.090 |
| ILPD | 0.909 | 81.869 | 0.325 | 23.254 | 0.004 | 0.097 |
| WDBC | 8.209 | 23.009 | 1.042 | 26.592 | 0.008 | 0.091 |
| pima | 1.580 | 66.515 | 0.471 | 54.542 | 0.005 | 0.105 |
| transfusion | 0.679 | 8.246 | 0.208 | 21.857 | 0.001 | 0.095 |
| banknote | 2.142 | 13.043 | 0.274 | 659.874 | 0.002 | 0.093 |
| mushroom | 1.637 | 16.369 | 2.083 | 48.763 | 0.031 | 0.252 |
| COMPAS-2016 | 2.860 | 14.914 | 1.243 | 33.815 | 0.003 | 0.159 |
| COMPAS-binary | 3.380 | 16.151 | 2.178 | 41.120 | 0.003 | 0.174 |
| FICO-binary | 7.705 | 11.199 | 6.890 | 72.515 | 0.016 | 0.432 |
| COMPAS | 16.534 | N/A | 10.359 | 237.615 | 0.083 | 0.897 |
| FICO | 33.935 | 159.838 | 18.826 | 695.484 | 0.215 | 1.121 |
| adult | 15.952 | 288.338 | 202.279 | 39787.330 | 0.815 | 4.802 |
| bank-market | 34.185 | 107.736 | 30.563 | 8956.680 | 0.124 | 0.842 |
| magic | 39.432 | 222.451 | 65.904 | N/A | 0.197 | 1.459 |
| musk | 88.215 | 659.791 | 371.562 | 864.823 | 1.388 | 1.644 |
| gas | 192.125 | 5353.880 | 582.690 | N/A | 2.331 | 2.772 |

## A.6 Approximation quality

On the first nine datasets in Table 6, all of the subproblems could be exactly solved within ten minutes using BnB. For the remaining higher-dimensional datasets, we specified a time limit of 10 minutes for BnB. Although the solutions obtained by time-limited search are not guaranteed to be close to the optimal ones, we noticed that our fine-tuned BnB procedure generally did not find a better solution beyond the first two minutes of its time limit, which indicates that nearly optimal solution had been reached (but had not been proved). The largest gap occurs on the COMPAS dataset, for which the BnB based method spent about ten hours in total, while the proposed method terminated in one minute.

Table 6: Approximation quality measured by relative gaps.

| Dataset | #features | $V(\mathcal{S}_{\text{approx}})$ | $V(\mathcal{S}_{\text{bnb}})$ | Relative Gap |
|---|---|---|---|---|
| COMPAS-binary | 24 | 871.00 | 875.00 | 0.0046 |
| COMPAS-2016 | 30 | 594.40 | 590.00 | -0.0075 |
| FICO-binary | 34 | 1977.00 | 1919.00 | -0.0302 |
| tic-tac-toe | 54 | 433.78 | 433.78 | 0.0000 |
| transfusion | 64 | 12.00 | 12.00 | 0.0000 |
| banknote | 72 | 599.40 | 602.40 | 0.0050 |
| heart | 118 | 99.48 | 99.48 | 0.0000 |
| ILPD | 160 | 217.00 | 217.00 | 0.0000 |
| mushroom | 224 | 3908.00 | 3908.00 | 0.0000 |
| liver | 104 | 127.68 | 124.69 | -0.0240 |
| pima | 134 | . 74.84 | 76.00 | . 0.0153 |
| bank-market | 174 | 3329.07 | 3323.59 | -0.0016 |
| magic | 180 | 9251.09 | 9193.73 | -0.0062 |
| COMPAS | 180 | 563.00 | 642.57 | 0.1238 |
| adult | 262 | 3690.00 | 3665.10 | -0.0068 |
| FICO | 312 | 1936.30 | 1927.00 | -0.0048 |
| WDBC | 540 | 209.00 | 207.01 | -0.0096 |
| ionosphere | 566 | 198.80 | 199.20 | 0.0020 |
| musk | 2922 | 565.50 | 609.90 | 0.0728 |
| gas | 2304 | 6234.64 | 6181.82 | -0.0085 |