# OpenReview forum: "Learning Interpretable Decision Rule Sets: A Submodular Optimization Approach"
_NeurIPS.cc/2021/Conference — NeurIPS 2021 Spotlight_

### Official Review · Reviewer_ijjJ · 2021-07-15

**Rating:** 7
**Confidence:** 5

**Summary:**

This submission addresses learning interpretable rule sets for binary classification problems. The approach is based on using submodular optimization, and the authors come up with a learning algorithm that uses approximation properties from DS optimization. The exact solution of the subproblems are NP-hard, and hence, the authors propose solving the subproblems via heuristic approaches. This is supported by local search operators. Some theoretical results are presented. The computational experiments show the effectiveness of their approach in terms of accuracy. The contribution of this work is an algorithm that works very well on binary classification and manages to find interpretable decision rules.

**Limitations And Societal Impact:**

The reasoning behind solving the subproblem via heuristic approaches is understandable. Nevertheless, the running times of the algorithms are not discussed/reported.

The experiments are given only for binary classification. The other missing point is details about generalization to multi-class classification.

**Main Review:**

Major Comments:
- How do you define small dataset and large dataset? Check lines 258, 262.
- Interpretability analysis: The number of rules and number of literals are significantly different than the
work by Dash et al. (2018) for CG and CART. Why? For RIPPER this is not the case and good. It would be
better to include the outcome of BRS in Table 2. Now the analysis looks incomplete.
- Limitations of this study are not discussed.

Minor Comments:
- 22-23: “Black box models are particularly problematic here because they are hard to audit and to diagnose.” => Sentence needs rewording, very informal.
- 77: one-versus-one or one-versus-all?
- 154: Missing a space after “cases,”.
- 218: “Decision lists are special cases of decision trees that grows only in one branching direction, and
the root-to-leaf paths of a decision tree form a rule set without overlap.” => Sentence needs rewording,
not clear.
- 220: “However, the learning of decision lists and trees are inherently much”
- 221: “harder because of their combinatorial structure is more complex than rule sets.”
- 251: “For BRS, the maximum”
- Very briefly, there are spelling mistakes in the document and the usage of language can be improved significantly...

**Time Spent Reviewing:**

6 hours

---

> ### Author Response · Authors · 2021-08-09
> **Response To Reviewer ijjJ**
>
> Thank you for the supportive reviews and valuable feedback! We are glad that you are positive about our paper. Below we address your concerns and questions in order.
>
> 1. RE: _How do you define small dataset and large dataset?_
>
> We divide these datasets roughly based on the running time of the proposed and baseline methods. We will revise the paper to make it more clear.
>
> 2. RE: _The number of rules and number of literals are significantly different than the work by Dash et al. (2018) for CG and CART. Why?_
>
> Thank you for pointing it out. The differences in the results for CG may be caused by the subproblem solving. Dash et al. (2018) reported results based on IP solver, while our reproduced results were obtained by running the official light reimplementation in which the subproblems are approximately solved with beam search. It seems that the beam search can only discover relatively simple rules, such that the reproduced complexities of rule sets are lower than Dash et al. (2018) on larger datasets.
>
> Regarding CART, through checking the experiment log, we find that outdated data of #literals were incorrectly reported in Table 2. The outdated data were obtained via running CART with default hyperparameter `min_samples_leaf=1`, which are inconsistent with the reported accuracy and #rules obtained via hyperparameter tunning. We are sorry about this mistake and will fix it in the revised paper. The correct data (retrieved from experiment log) are:
>
> ```csv
> dataset,          #Literals
> tic-tac-toe,    138.8 (7.1)
> liver,            9.0 (0.0)
> heart,           21.8 (2.1)
> ionosphere,      48.4 (4.2)
> ILPD,             7.6 (1.0)
> WDBC,            14.8 (2.0)
> pima,            19.2 (1.1)
> transfusion,     21.0 (0.9)
> banknote,        55.4 (5.4)
> mushroom,        27.2 (1.1)
> COMPAS-2016,     61.0 (1.3)
> COMPAS-binary,  155.0 (2.1)
> FICO-binary,    316.0 (5.4)
> COMPAS,         170.8 (4.7)
> FICO,           138.0 (2.7)
> adult,          795.6 (9.9)
> bank-market,    577.0 (5.7)
> magic,          796.8 (9.3)
> musk,          359.0 (14.2)
> gas,            343.4 (4.7)
> ```
>
> These values are usually smaller than their incorrectly reported counterparts and more comparable with Dash, et al. (2018) for smaller datasets.
>
> 3. RE: _Limitations of this study are not discussed_
>
> A more detailed discussion on the limitations of the proposed method is added in the overall response.
>
> 4. RE: _Writing issues and typos_
>
> Thank you very much for pointing out these issues. We will fix them in the revised paper.
>
> 5. RE: _Running times of the algorithms_
>
> The running times of the methods are now reported and analyzed in the overall response.
>
> 6. RE: _Multi-class classification_
>
> We add a detailed discussion on the extension to multiclass classification in the overall response.

---

### Official Review · Reviewer_4Q8k · 2021-07-16

**Rating:** 6
**Confidence:** 4

**Summary:**

This paper proposes an approach for learning decision rule sets that combines both rule generatino and rule selection. this is accomplished by formulating the an outer subproblem as regularized submodular maximization and an inner problem as maximizing the difference of two submodular functions. Experimental results show that the method has improved test performance with less overlap among decision rules.


**Limitations And Societal Impact:**

No explicit discussion of limitations or negative social impact. The authors could discuss that in general interpretable classification methods need not be more fair or accurate than large neural networks.

**Main Review:**

Originality: Interesting but straightforward combination of previous work on submodular maximization (distorted greedy, majorize-maximization) to a new domain.

Quality:
- The authors should discuss how the rule set compares to the true optimal rule set, since two solutions with similar values could be differ in their interpretability.
- While test accuracy is impressive (Table 1), the results in Table 2 are much more impressive for Overlap percentage, which makes sense because the proposed algorithm is admittedly the only one to minimize that explicitly (line 276). Since #Rules and #Literals are often much worse than baselines, it would be good to argue strongly that overlap is a much more important interpretability measure than either of those.

Clarity: Paper is clear, well-organized, and generally a pleasure to read

Methodology:
- Why set $\beta_o = \beta_1 = 1$? Why fix the number of trees in the RF to be 100?
- What was the problem with running [27]? Since references [14, 27] are discussed in related work and appear to be very relevant, the paper would be improved if it added these works as baselines.
- Figure 1 (scalability) should include other algorithms as well

Quality/Significance: If the inner subproblem is solved with an approximate local search heuristic, then the theoretical guarantee of Proposition 2, which assumes the inner subproblem is solved to exact optimality, is not relevant. The paper does not address this fully, and would benefit from identifying how much is lost to this approximate inner subproblem either theoretically (assume an approximate inner maximization oracle) or empirically (ablation on different inner subproblem methods).

Questions:
- Do other approaches require binarizing the features? If not, how do they scale with d?

---
EDIT: The authors addressed my main concerns with additional experiments showing the optimality gap with respect to a branch and bound method. I am increasing my score to 6, although the paper would be further strengthened by studying the sensitivity to inner subproblem solver in even greater detail.


**Time Spent Reviewing:**

4

---

> ### Author Response · Authors · 2021-08-09
> **Response To Reviewer 4Q8k**
>
> Thank you for the detailed review and constructive feedback. We address your concerns in order.
>
> 1. RE: _The authors should discuss how the rule set compares to the true optimal rule set_
>
> Unfortunately, the true optimal rule sets for these datasets are hard to obtain, except the tic-tac-toe dataset for which we know the golden solution according to the rules of this game. As a surrogate, we conducted a comparison to exact subproblem solving on several tractable low-dimensional datasets. According to Proposition 2, the quality of a solution obtained by exact subproblem solving is close to the optimal solution. The results are reported and analyzed in the overall response.
>
> 2. RE: _It would be good to argue strongly that overlap is a much more important interpretability measure_
>
> Thanks for your suggestion. Small overlap is particularly beneficial to instance-wise explanation, i.e., explaining why a specific example is assigned a positive label. The example will be matched by only a few (usually one) rules in a rule set with small overlap, and thus the generated explanation will be simple and easy to understand. We will add this argument to a new version of the paper.
>
> 3. RE: _Why set $\beta_0 = \beta_1 = 1$?_
>
> These two hyperparameters are fixed to make the experimental comparison with CG relatively fair. In the appendix, we have shown that the Hamming loss in CG is equal to $L(\mathcal{S})$ with $\beta_0 = \beta_1 = 1$, $\beta_2 = 0$ and $\lambda = 0$.
>
> 4. RE: _Why fix the number of trees in the RF to be 100?_
>
> This hyperparameter defaults to 100 in scikit-learn. We did not observe significant improvements in accuracy by setting it to larger values.
>
> 5. RE: _What was the problem with running [27]?_
>
> We failed to run publicly available implementations of IDS on most of the datasets within practical running time, which is consistent with the experience of Dash, et al. [14, Section A.4].
>
> 6. RE: _Figure 1 (scalability) should include other algorithms as well_
>
> Thanks for your suggestion. The running times of the methods are now reported and analyzed in the overall response. We would like to add similar scalability tests for baselines in the appendix of the revised paper.
>
> 7. RE: _Do other approaches require binarizing the features? If not, how do they scale with d?_
>
> Binarization is not required for CART and RF, as their built-in tree splitting procedure can handle mixed-type data natively. However, all datasets were still binarized for CART and RF in our experiments to make the comparison fair. The time complexity for the growing of a decision tree typically scales linearly with the number of features.
>
> 8. RE: _No explicit discussion of limitations or negative social impact_
>
> Following your suggestion, a detailed discussion on the limitations and societal impact of the proposed method is added in the overall response.
>
> We hope our response resolves your main concerns and would be grateful if you could increase the rating.

---

### Official Review · Reviewer_Uvxy · 2021-07-21

**Rating:** 7
**Confidence:** 3

**Summary:**

This paper addresses learning rule set models for binary classification, ie combinations of logical rules on features to classify a positive class. Since learning rule sets requires selection of features over which the logical rules are applied and selection of the rules themselves, the subset selection problem is exponential. This paper formulates the task loss as a supermodular function comprised of a constant term (ignored in the optimization), a modular function term and a submodular function term. The paper outlines an optimization algorithm that contains a subproblem that is not computationally feasible except for small problems. Thus, it outlines an approximate solution to the subproblem.

**Limitations And Societal Impact:**

Limitations of the method are not extensively addressed, especially since the approximate solution loses the approximation guarantees admitted by exactly solving the marginal gain maximization subproblem, which can only be computed for small problems. For a small problem like COMPAS-binary with 24 features or other small datasets, the solution could be compared to the exact subproblem solving both in accuracy and running time to investigate some of the gaps in the approximation.

**Main Review:**

The main contribution of the work is in the specific formulation of the rule set learning problem as a regularized submodular loss function and then applying the optimization algorithm of [2] to solve it. However the subproblem introduced in that algorithm cannot be solved exactly for larger problems, so the paper proposes an approximate solution to the difference of submodular functions. Thus, the optimization does not have any approximation guarantees. The method may be useful in it's formulation, though the optimization of the loss is less compelling without real 'nice' properties beyond the results that can be seen empirically, ie it seems to work in practice. Without more data on running time, both on the proposed optimization and the baselines, the empirical analysis is still lacking.

Clarity: The work is generally well presented and the paper is mostly clear. One exception is the discussion of related work. Section 4 does not go into enough detail of the cited work to place this work into the context of the field. The introduction section even provides more information on some of these works. Can these be combined or reorganized to better contextualize the contribution of the current work as well as the existing work of the relevant literature that is already cited?
Experiments: It’s not clear why an alternative (light) implementation of the Column Generation (CG) [1] method was used. While the beam search alternative may be more comparable in that it is an approximate algorithm, the full method can be evaluated against the proposed method both on accuracy and speed, for example. Is a full implementation not available? In any case, the running time of the proposed and baseline methods should be reported as well, especially since some methods were excluded based on a running-time cut-off.
[1] Harshaw, C., Feldman, M., Ward, J. and Karbasi, A. "Boolean decision rules via column generation." Proceedings of the 32nd International Conference on Neural Information Processing Systems. 2018.
[2] Harshaw, Chris, et al. "Submodular maximization beyond non-negativity: Guarantees, fast algorithms, and applications." International Conference on Machine Learning. PMLR, 2019.



Post rebuttal comments:
The new results and clarifications from the authors have strengthened the experimental components of the submission, which I believe are the main contribution of the work. Thus, I am updating my score to recommend acceptance.

**Time Spent Reviewing:**

5

---

> ### Author Response · Authors · 2021-08-09
> **Response To Reviewer Uvxy**
>
> Thanks for your constructive feedback. Below we summarize your questions and address them in order.
>
> 1. RE: _Running time of the proposed and baseline methods_
>
> Following your suggestion, the running times of the methods are reported and analyzed in the overall response.
>
> 2. RE: _Organization of related work_
>
> Thank you for pointing it out. In a new version of the paper, we will reorganize the introduction section and the related work section to put details into the latter.
>
> 3. RE: _Why a light implementation of CG was used_
>
> An implementation based on IP solver is not publicly available, so we choose the official light edition implemented in the AIX360 toolkit by IBM Research.
>
> 4. RE: _Limitations of the method are not extensively addressed, especially the loss of approximation guarantees_
>
> Following your suggestion, we compared to exact subproblem solving on several tractable datasets. The results are reported and discussed in the overall response. We also add the discussion on other limitations of the proposed method there.
>
> We hope our response resolves your main concerns and would be grateful if you could increase the rating.

---

### Author Response · Authors · 2021-08-09
**Overall Response**

We sincerely thank the reviewers for their time and appreciate all the detailed reviews and constructive feedback. Extra experimental results and discussion are presented below to address some common concerns raised by reviewers. In addition, each review will be replied to individually. The discussion here will be properly incorporated into a new version of our paper (mostly in the appendix, more specifically).

## Running Time

Following the suggestion by all reviewers, the average running time in seconds of each method for fitting each dataset was measured on a MacBook Pro (2019 edition, Intel Core i5). The hyperparameters of the methods were set to their typical values found in cross-validation. Results are summarized in the following table.

```csv
dataset,         ours,        CG,   RIPPER,        BRS,   CART,     RF
tic-tac-toe,    0.794,    12.815,    0.204,     14.833,  0.002,  0.097
liver,          4.113,    62.482,    0.232,     19.513,  0.002,  0.083
heart,          0.853,    62.840,    0.227,     14.858,  0.001,  0.077
ionosphere,     6.064,    51.475,    0.914,     17.304,  0.008,  0.090
ILPD,           0.909,    81.869,    0.325,     23.254,  0.004,  0.097
WDBC,           8.209,    23.009,    1.042,     26.592,  0.008,  0.091
pima,           1.580,    66.515,    0.471,     54.542,  0.005,  0.105
transfusion,    0.679,     8.246,    0.208,     21.857,  0.001,  0.095
banknote,       2.142,    13.043,    0.274,    659.874,  0.002,  0.093
mushroom,       1.637,    16.369,    2.083,     48.763,  0.031,  0.252
COMPAS-2016,    2.860,    14.914,    1.243,     33.815,  0.003,  0.159
COMPAS-binary,  3.380,    16.151,    2.178,     41.120,  0.003,  0.174
FICO-binary,    7.705,    11.199,    6.890,     72.515,  0.016,  0.432
COMPAS,        16.534,       N/A,   10.359,    237.615,  0.083,  0.897
FICO,          33.935,   159.838,   18.826,    695.484,  0.215,  1.121
adult,         15.952,   288.338,  202.279,  39787.330,  0.815,  4.802
bank-market,   34.185,   107.736,   30.563,   8956.680,  0.124,  0.842
magic,         39.432,   222.451,   65.904,        N/A,  0.197,  1.459
musk,          88.215,   659.791,  371.562,    864.823,  1.388,  1.644
gas,          192.125,  5353.880,  582.690,        N/A,  2.331,  2.772
```

First of all, the readers should be aware that the running time comparison based on wall-clock time here is not totally fair, as the implementations of these methods are not optimized to a matching level. Among the baselines, CG, BRS and RIPPER are implemented in Python (with numerical computation offloaded to numpy), while CART and RF are based on a highly optimized Cython backend. Our method is implemented in Golang, a programming language that in general is faster than Python and is slower than C/C++. Nevertheless, this table, jointly with the reported experimental data on accuracy and interpretability, enables us to roughly understand the trade-offs of these methods.
- RF is fast and accurate, yet not interpretable.
- RIPPER is also fast, but it usually produces more complex rule sets than our approach.
- BRS is relatively slow due to stochastic search and is less accurate than the remaining methods.
- The CG method produces the simplest rule sets on many datasets. However, despite solving the subproblem with beam search rather than IP solver, it is still time-consuming. Moreover, it is less accurate than the proposed method on larger datasets.

Generally speaking, our method strikes a good balance among accuracy, interpretability and scalability.

## Approximation Quality

Following the suggestion by Reviewer Uvxy and Reviewer 4Q8k, we conducted a comparison between approximate subproblem solving and exact subproblem solving. We found that on nine low-dimensional datasets, all of the subproblems could be exactly solved within ten minutes using our customized branch-and-bound procedure. For each of these datasets, we recorded the values of the objective function $V(\mathcal{S})$ achieved by the learned rule sets under the two setups (using the same typical hyperparameters). Results are summarized in the following table.

```csv
dataset,        #dim,  obj (approx),  obj (exact),      gap
COMPAS-binary,    24,        871.00,       875.00,   0.0046
COMPAS-2016,      30,        594.40,       590.00,  -0.0075
FICO-binary,      34,       1977.00,      1919.00,  -0.0302
tic-tac-toe,      54,        433.78,       433.78,   0.0000
transfusion,      64,         12.00,        12.00,   0.0000
banknote,         72,        599.40,       602.40,   0.0050
heart,           118,         99.48,        99.48,   0.0000
ILPD,            160,        217.00,       217.00,   0.0000
mushroom,        224,       3908.00,      3908.00,   0.0000
```

The gaps are the relative differences between the solutions produced via approximate and exact subproblem solving, calculated by $[ V(\mathcal{S}\_{\text{exact}}) - V(\mathcal{S}\_{\text{approx}})] / V(\mathcal{S}\_{\text{exact}})$. These small gaps indicate that the proposed approximate algorithm works quite well on these datasets. For instance, the two setups produced exactly the same rule sets on five of the datasets. Interestingly, the cases of COMPAS-2016 and FICO-binary illustrate that approximate subproblem solving is not necessarily worse than the exact counterpart. This is because the outer loop (Algorithm 1) is also an approximation procedure, which does not guarantee that better subproblem solutions will help build a better solution to the main problem.


## Limitations

### Approximation Guarantee

The main limitation of this work is the invalidity of approximation guarantee (Proposition 2) under inexact subproblem solving, which we have explicitly pointed out in the paper (Line 154). Nonetheless, the approximation guarantee is still relevant in the case that the optimal solution to the subproblem can be found. This is possible if we restrict the space of rules to a tractable extent. For example, if we move back to the pre-mining approach in which rules can only be picked from a given pool (e.g., rules extracted from a random forest), the guarantee will still hold for this modified problem. Furthermore, the previous discussion of approximation quality shows that empirically such invalidity will not necessarily lead to much loss of performance. We are planning to analyze this phenomenon theoretically in future work (through assuming an approximate inner maximization oracle as suggested by Reviewer 4Q8k, for example).

### Multiclass Extension

Another more subtle limitation mentioned by Reviewer ijjJ is the extension to multiclass classification. We have suggested the model-agnostic one-versus-one (OvO) transformation in the paper (Line 77), which only requires the base classifiers to produce a binary class label, rather than a real-valued confidence score required by the one-versus-rest (OvR) approach. However, both OvR and OvO may suffer from ambiguities and class imbalance. Moreover, the OvO reduction is not very efficient because $K(K-1)/2$ base classifiers must be trained for a $K$-class classification task. We are investigating if there is a more scalable approach to extend this work to multiclass problems.


## Societal Impact

On the positive side, the underlying decision logic of an interpretable model such as a rule set is more transparent for humans to understand. When used as a data exploration tool, such transparency enables people to check easily if the fitted model has discovered some new knowledge about the data. When deployed to make automatic decisions in high-stake scenarios, such transparency helps people to diagnose if there are potential biases or risks in the model and to explain the reason behind a specific decision.

On the negative side, as pointed out by Reviewer 4Q8k, a rule set learned by optimizing accuracy and interpretability will not automatically attain other properties requested by trustworthy AI, such as fairness and robustness. In addition, a learned rule ```IF X=x0 AND Y=y1 THEN Z=1``` may not conform to the causal mechanism underlying the variables X, Y and Z. This will lead to misunderstandings if the rules are not interpreted properly.

---

> ### Comment · Reviewer_4Q8k · 2021-08-26
> **Approximation Quality for Remaining Datasets**
>
> Thank you for the detailed, informative response. Regarding "Approximation Quality", is it possible to show the gap on all datasets, not just the low dimensional ones? While your initial experiment shows the relative gap between approximate and exact subproblem is quite small, it's possible that this gap grows with the dimensionality (or size) of the problem instance. Also, running additional maximization methods would show how sensitive the proposed approach is to the choice of inner subproblem solver.

---

> > ### Author Response · Authors · 2021-08-29
> > **RE: Approximation Quality for Remaining Datasets**
> >
> > Thank you very much for participating in discussions! As the subproblems for remaining higher-dimensional datasets could not be solved to optimality with practical running time, we specified a time limit of 10 minutes for the BnB subproblem solver and collected the achieved objective function values. The results are summarized in the following table.
> >
> > ```csv
> > dataset,        #dim,  obj (approx),  obj (bnb-10min),       gap
> > bank-market,     174,       3329.07,          3323.59,   -0.0016
> > magic,           180,       9251.09,          9193.73,   -0.0062
> > COMPAS,          180,        563.00,           642.57,    0.1238
> > adult,           262,       3690.00,          3665.10,   -0.0068
> > FICO,            312,       1936.30,          1927.00,   -0.0048
> > WDBC,            540,        209.00,           207.01,   -0.0096
> > ionosphere,      566,        198.80,           199.20,    0.0020
> > musk,           2922,        565.50,           609.90,    0.0728
> > gas,            2304,       6234.64,          6181.82,   -0.0085
> > ```
> >
> > It can be observed that the relative gaps are still small in general and do not grow with the dimensionality. Although the solutions obtained by time-limited search are not guaranteed to be close to the optimal ones, we noticed that our fine-tuned BnB procedure generally did not find a better solution beyond the first two minutes of its time limit, which indicates that nearly optimal solution had been reached (but had not been proved). The largest gap occurs on the COMPAS dataset, for which the BnB based method spent about ten hours in total, while the proposed method terminated in one minute.

---

### Decision · Program_Chairs · 2021-09-27

**Decision:**

Accept (Spotlight)

**Comment:**

The authors provide an interesting connection between an interpretable decision rule set and submodular optimization. Technically, they either consider regularized submodular maximization framework or the difference of two submodular functions.

I found the formalization very interesting. Also, the authors had a successful rebuttal and convincing new set of experiments.  Regarding scaling up the experiments, the authors may look at a recent ICML paper on "Regularized Submodular Maximization at Scale" by Kazemi et al. Also, there is a related paper on interpretability using submodularity that the authors may consider looking into "Streaming Weak Submodularity: Interpreting Neural Networks on the Fly" by Elenberg et al.

All in all, this is an interesting paper and I suggest acceptance.